# Robust Reinforcement Learning for Portfolio Optimization via Cooperation and Competition Strategies

## Abstract

In this study, we propose an intelligent system for portfolio management that applies robust reinforcement learning within a multi-agent framework. The proposed system incorporates both competition and cooperation strategies to enhance decision-making performance and adaptability. By formulating the portfolio management problem as a cooperative multi-agent environment, agents collaborate and jointly strive to achieve a common goal. On the other hand, the inclusion of competition strategies enables agents to dynamically compete for limited resources and advantages in the market. Specifically, the proposed cooperative strategies employ the absolute value of the reward, prioritizing accelerated model convergence. Meanwhile, the competitive strategies utilize previous rewards to guide action selection, aiming to seek gains and avoid losses. To assess the performance of our model, we evaluate it on a set of real-world financial data. The results obtained demonstrate that the proposed game strategies outperform traditional reinforcement learning approaches.

## 1 Introduction

Portfolio management has always been a challenging and dynamic field, demanding traders to make informed decisions amidst rapidly changing market conditions. As the volume and complexity of financial data continue to increase, there is a growing interest in utilizing machine learning techniques to address this problem. Specifically, machine learning has emerged as a promising avenue for developing automated investment strategies that can adapt to real-time market changes Choi et al. (2009); Leippold et al. (2022). Therefore, the main objective of this study is to explore the application of an intelligent system to enhance the robustness of portfolio optimization strategies.

Reinforcement Learning (RL) is a machine learning technique that empowers an agent to acquire decision-making capabilities through interaction with its environment and receiving feedback in the form of rewards or penalties. In the realm of quantitative trading, RL has brought about a revolution by enabling traders to make well-informed decisions based on historical market data and adapt their strategies in response to market dynamics (Wang et al., 2021). However, despite the evident benefits of RL, there are potential drawbacks that require attention. For instance, RL-based trading strategies can be highly sensitive to changes in the environment, where even minor variations can result in a significant decrease in performance or, in some cases, catastrophic failure (Tessler et al., 2019). To address this challenge, this paper focuses on developing a robust RL model for portfolio management using competition and cooperation strategies within a multi-agent system, referred to as CCRRL (Robust Reinforcement Learning via Multi-Agent System).

The main contributions of this work are based on the following features:

**1)** It introduces a robust reinforcement learning architecture based on multi-agent system, which aims to mitigate the negative impact of variations in the market, ensuring the stability and adaptability of the portfolio management system.

**2)** It incorporates both competition and cooperation strategies within the multi-agent system, leading to collective enhancements in the performance of reinforcement learning algorithms.

**3)** It performs extensive experimental evaluations and comparisons, where the experiments involve diverse market scenarios and various investment strategies to ensure the validity and generalizability of the findings.

## 2    RELATED WORKS

The integration of RL into portfolio optimization represents a significant breakthrough in the advancement of trading strategies (Zhang et al., 2020; Liu et al., 2022) . Recently, Wang et al. (2021) introduced a deep RL framework to enhance portfolio selection strategies by applying the Tucker decomposition technique to solve multimodal problems. However, the real world is filled with uncertainties and disturbances that can cause RL agents to fail or underperform. This is where the concept of Robust Reinforcement Learning (RoRL) becomes relevant. RoRL is a subfield of RL that aims to enhance the stability, reliability, and generalization of RL algorithms in the face of various disturbances and uncertainties (Pinto et al., 2017; Kamalaruban et al., 2020). The primary objective of RoRL is to empower RL agents to learn policies capable of adapting to unexpected changes and recovering from failures in uncertain and dynamic environments. To achieve this objective, researchers have proposed various methods for robust RL under uncertain and dynamic environments, including model-based uncertainty estimation (Osband et al., 2018; Lütjens et al., 2019), RL with auxiliary tasks (Hernandez-Leal et al., 2019; Lin et al., 2019), and transfer learning (Zhan & Taylor, 2015; Shi et al., 2021).

In the realm of RL with auxiliary tasks, the utilization of multi-agent game theory has garnered significant attention for enhancing the robustness of RL algorithms. Kartal et al. (2019) proposed a novel defense strategy based on time-varying channels, treating the process of countering malicious interference as a multi-user intelligent game model. By considering the unknown interference model and strategy, their work introduced a multi-user random post-decision state game algorithm that employs deep reinforcement learning to intelligently fend off intelligent attackers. Additionally, Jiang et al. (2022) addressed a crucial issue in the stock market: maximizing returns by selling stocks at the opportune moment. The authors emphasized the influence of human emotions and greed on investors' selling decisions and stressed the importance of developing rational strategies to generate investment returns. To tackle this challenge, they constructed a multi-agent model using deep reinforcement learning, which outperformed other machine learning models by adapting better to complex market changes and facilitating superior selling decisions for investors. Therefore, the central focus of this paper revolves around the development of a Robust Reinforcement Learning via Multi-Agent System (CCRRL) model explicitly designed to fortify the robustness of portfolio strategies. By drawing inspiration from the advancements in multi-agent game theory and integrating robust reinforcement learning techniques, this study aims to enhance the stability and adaptability of portfolio management strategies in the face of uncertainties and disturbances in financial markets.

Cooperative reinforcement learning represents an extended approach to the traditional reinforcement learning framework, specifically addressing collaboration problems in multi-agent systems (Shi, 2020). It involves the mutual cooperation and collaboration among multiple agents to achieve common goals. Lee et al. (2020) introduced a multi-agent reinforcement learning network for generating investment strategies in stock markets, applying advanced deep learning methods. In their approach, each agent operates independently and manages its own portfolio. The results indicate that the inclusion of more agents in the system further improves the Sharpe ratio by reducing risk through a more diversified portfolio. For further details on cooperative reinforcement learning, interested readers may refer to the following papers (Foerster et al., 2016; Liu et al., 2018; Lowe et al., 2017). On the other hand, competitive reinforcement learning extends the classical framework of reinforcement learning to scenarios where multiple agents compete against each other. Daskalakis et al. (2020) have obtained global, non-asymptotic convergence guarantees for independent learning algorithms in competitive reinforcement learning settings with two agents. Their work demonstrates that if both players simultaneously employ policy gradient methods and their learning rates follow a two-timescale rule, their policies will converge to a minimax equilibrium of the game. For further insights into competitive reinforcement learning, interested readers may refer to the following papers (Ye et al., 2020; Bai & Jin, 2020).

## 3 METHODOLOGY

### 3.1 VARIABLES DESCRIPTION

The principal formulas employed in this article are delineated in Table 1. To cater to distinct objectives, we partition the variables into two distinct groups. The first group serves the purpose of constructing a mathematical programming model for the portfolio optimization, while the second group expounds upon the RL model utilized to solve the portfolio optimization problem.

| Notations | Descriptions |
|---|---|
| | Mathematical Programming |
| $x_{it}$ | The proportion of the total investment allocated to asset $i$ at time step $t$. |
| $r_{it}$ | The expected return of asset $i$ at time step $t$. |
| $\sigma_i$ | The standard deviation (or volatility) of asset $i$. |
| $tr$ | The desired target return for the portfolio. |
| $\rho_{ij}$ | The correlation coefficient between assets $i$ and $j$. |
| | Reinforcement Learning |
| $s_t$ | The current state at time step $t$. |
| $action_t$ | The action generated by an agent at time step $t$. |
| $reward$ | The previous reward of an agent. |
| $\mu$ | The deterministic strategy function. |
| $\theta_\mu$ | The parameter of deterministic strategy function. |
| $\mathcal{N}$ | The random noise. |
| $action_{adv}$ | The action of a competitive agent. |
| $action_{cop}$ | The action of a cooperative agent. |
| $action_{pre}$ | The previous action of an agent. |
| $action_{fin}$ | The output of the final action. |
| $modify$ | The modification function adjusting the reward. |

Table 1: Descriptions of main considered variables.

### 3.2 MODELS DESCRIPTION

#### 3.2.1 MATHEMATICAL PROGRAM FOR THE PO

A standard mathematical program for the dynamic portfolio optimization problem can be written as follow.

$$(\text{P}) \qquad Maximize \quad z = \sum_{i=1}^{n} r_{it} x_{it} \tag{1}$$

$$\text{s.c.} \quad \sum_{i=1}^{n} x_{it} = 1, \forall t = 1, \ldots, T, \tag{2}$$

$$\sum_{i=1}^{n} r_{it} x_{it} \geq tr, \forall t = 1, \ldots, T, \tag{3}$$

$$\sum_{i=1}^{n} \sum_{j=1}^{n} \sigma_i \sigma_j \rho_{ij} x_{it} x_{jt} \leq \sigma_{\max}^2, \tag{4}$$

$$x_{it} \geq 0, \forall i = 1, \ldots, n, \ \forall t = 1, \ldots, T.$$

In this model, the decision variable $x_{it}$ represents the proportion of the total investment allocated to asset $i$ at date $t$, for all $i = 1, 2, \ldots, n$. The objective function (Equation 1) is to maximize the expected return of the portfolio $z$, which is the sum of the expected returns of the individual assets weighted by their respective proportions. The budget constraint (Equation 2) ensures the total allocation sums to 1, while the target return constraint (Inequality 3) ensures the portfolio's expected return is greater than or equal to the desired target return $tr$. Additionally, there is a risk constraint (Inequality 4), often represented by the portfolio variance, that limits the overall risk of

the portfolio. Inequality 4 ensures that the portfolio's variance, calculated as the weighted sum of the asset variances and covariances, is less than or equal to a specified maximum variance $\sigma_{\max}^2$. The correlation coefficients $\rho_{ij}$ capture the relationship between the returns of different assets and are used in calculating the portfolio variance. The solution to P provides the optimal allocation of funds to each asset in the portfolio, considering the desired target return and the risk constraints. It allows investors to construct diversified portfolios that balance risk and return, maximizing the expected return while staying within acceptable risk limits.

### 3.2.2 SOLUTION METHOD

Within the framework of CCRRL proposed for portfolio optimization, a key component is the integration of the Deep Deterministic Policy Gradient (DDPG) algorithm Lillicrap et al. (2015). DDPG is a promising RL technique that utilizes a Deep Neural Network (DNN) to learn an optimal policy for tasks involving continuous control. By combining the strengths of Q-learning and policy gradients, DDPG effectively learns a policy that maps states to actions in a given environment. Notably, DDPG applies the power of the DNN to represent both the policy and the Q-value function, enabling the handling of high-dimensional state and action spaces. The policy network maps the current state to an action, while the Q-network estimates the expected cumulative reward for a specific state-action pair.

However, factors such as network structure and random parameter initialization introduce unpredictability in the output of the Actor network, even for the same input state. This inherent randomness can significantly impact the training process, leading to challenges in convergence, learning, and overall instability. To address these instability issues, we propose the CCRRL model. Unlike conventional DDPG algorithms, CCRRL incorporates principles from game theory and considers interactions with other agents during the training phase. In this context, the actor network must account for its interaction with other agents and continuously adapt its strategy to optimize reward attainment. By incorporating competition and cooperation strategies into the training process, CCRRL enhances stability and convergence capabilities. It enables the actor network to dynamically adjust its actions based on the actions of other agents, resulting in more robust and effective learning. As a result, CCRRL offers an improved approach to tackle the challenges posed by randomness and instability in traditional DDPG algorithms.

### 3.3 OVERVIEW OF CCRRL

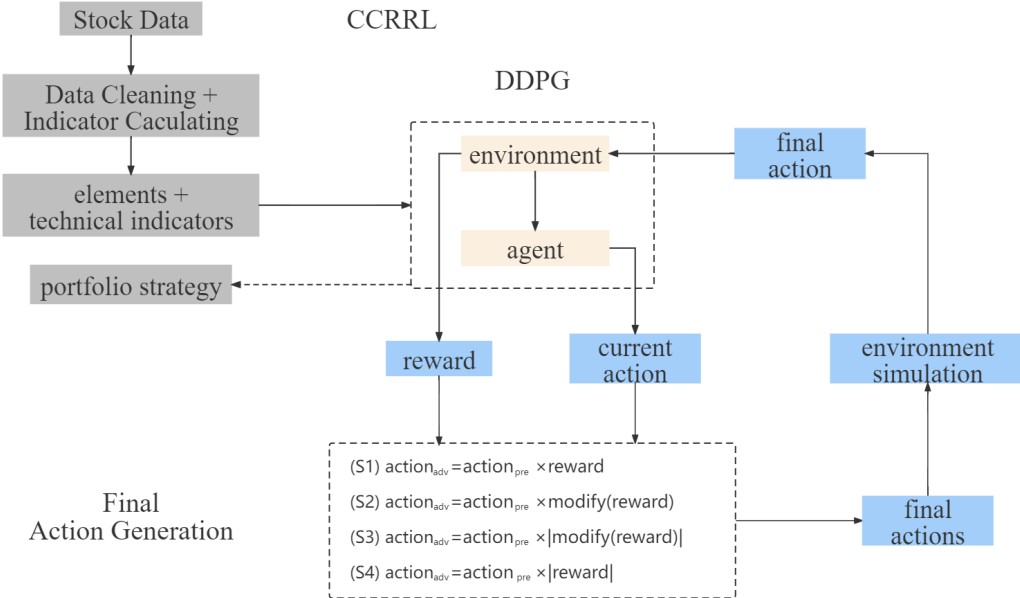

Figure 1: The main concept of CCRRL.

Figure 1 illustrates the key concept of the proposed CCRRL, which aims to enable robust portfolio selection strategies through RL. The framework consists of interconnected components that collaborate to generate investment actions. In CCRRL, the DDPG algorithm acts as the primary agent responsible for generating actions based on the current state of the stock market. The DDPG model takes stock data as input and produces both a reward signal and an initial action based on the market's current state. Once the DDPG model generates an action, it undergoes processing in the $modify$ function. The reward signal is modified, and the resulting value is multiplied by the initial action to derive the adversary action $action_{adv}$. $action_{adv}$ is then added to the original action, resulting in the final action $action_{fin}$ that is executed in the market. Then put these four final actions into environment simulation in order to find which final action could generate the best reward. After each simulation, the environment will be reset to the state before simulation. After generating $action_{fin}$, it is returned to the DDPG model, which generates corresponding rewards and uses $action_{fin}$ as $action_{pre}$ for the next training step. This process of action generation and execution continues in a loop until the training is completed.

It is important to note that the concept of $reward$ is fundamental in RL as it transforms tasks into a series of feedback signals that facilitate effective communication between the agent and the environment. In CCRRL, $reward$ represents the value obtained by an agent through the selection of an action, serving as a measure of that action's contribution to the task objective. Typically, $reward$ is represented as a real value, and the main objective of CCRRL is to accomplish tasks by selecting a series of continuous actions that maximize cumulative rewards. Therefore, we are developing a multi-agent game mechanism centered around the concept of $reward$ to enhance the robustness of the reinforcement learning model.

## 4 MULTI-AGENT SYSTEM VIA COMPETITION AND COOPERATION STRATEGIES

### 4.1 AGENT'S PREVIOUS ACTION

Recall that $action_t$ is utilized by the DDPG algorithm to generate actions based on the current state $t$ of the system. The mathematical representation of $action_t$ is provided in Equation 5.

$$action_t = \mu(s_t \mid \theta_\mu) + \mathcal{N}, \tag{5}$$

Here, $\mu$ represents a deterministic strategy function that maps the current state $s_t$, while $\theta_\mu$ denotes the parameter vector governing the behaviour of this deterministic strategy function. Additionally, $\mathcal{N}$ refers to the random noise incorporated into the function's output to facilitate exploration and diversification.

In the context of CCRRL, $action_{pre}$ is employed to represent the action generated in the previous step. Notably, $action_{pre}$ plays a significant role in the generation of $action_{adv}$ in the current step. By applying the previous action, CCRRL incorporates temporal information and adapts its decision-making process accordingly. This integration of temporal information enables CCRRL to make more informed and contextually appropriate decisions.

### 4.2 AGENT'S ACTION

Competition refers to the behavioural and strategic confrontation between two or more individuals, organizations, or teams as they compete for limited resources or advantages. Conversely, cooperation entails the behavioural and strategic collaboration of two or more individuals, organizations, or teams, working collectively toward a shared objective. Based on the aforementioned concepts, Strategies S1 and S2 can be categorized as competitive strategies, while Strategies S3 and S4 can be classified as cooperative strategies. The rationale for this classification is as follows: Strategies S3 and S4 adopt the absolute value of the reward and do not consider the potential profitability of the current action. Their primary objective is to expedite the convergence of the model, making them cooperative strategies. In contrast, Strategies S1 and S2 do not utilize the absolute value of the reward when selecting actions. If the preceding reward is negative, the generated adversarial action will have the opposite sign to the previous action. Conversely, if the preceding reward is positive, the generated adversarial action will align with the same sign as the previous action, ultimately converging towards the direction of the previous action value. This approach aims to optimize gains

and mitigate losses, positioning them as competitive strategies. he main subjects of cooperation and competition are four models trained with s1-s4. There is no data interaction between them. They just select the best model from them in each round and provide actions to train ddpg in the simulation. However, they share a environment, there is indirect data interaction.Since the four models do not affect the environment at the same time, they can cope with the problem of unstable environment.

Opponents are virtualized using S1-S4, and its significance is to use the results of the previous step to guide the action generation direction of this step.

### 4.2.1 COMPETITIVE AGENT'S ACTION

$action_{adv}$ represents the action taken by the opponent in the current state. This opponent is generated by the CCRRL model based on the output of the previous step. By considering the opponent's action, CCRRL enhances its anti-interference ability and adaptability, enabling it to effectively handle external disturbances. We present five adversary strategies that employ different approaches for generating the opponent's action.

In the first strategy, denoted as $S1$, $action_{adv}$ is obtained by multiplying $action_{pre}$ and $reward$, resulting in the desired $action_{adv}$. The mathematical representation of $action_{adv}$ is given by Equation 6:

$$(\textbf{S1}) \quad action_{adv} = action_{pre} \times reward \tag{6}$$

where $action_{pre}$ is the action generated in the previous step of CCRRL.

In the second strategy, denoted as $S2$, $action_{adv}$ is obtained by multiplying $action_{pre}$ and the absolute value of $reward$ that has been processed by the modify function, resulting in the desired $action_{adv}$. The mathematical representation of $action_{adv}$ is given by Equation 7:

$$(\textbf{S2}) \quad action_{adv} = action_{pre} \times \text{modify}(reward) \tag{7}$$

where $action_{pre}$ is the action generated in the previous step of CCRRL, and modify represents the adjustment of the range of values for the rewards.

### 4.3 COOPERATIVE AGENT'S ACTION

In the third strategy, denoted as $S3$, $action_{adv}$ is generated by multiplying $action_{pre}$ and the absolute value of $reward$ that has been processed by the modify function. The mathematical representation of $action_{adv}$ is given by Equation 8:

$$(\textbf{S3}) \quad action_{adv} = action_{pre} \times |\text{modify}(reward)| \tag{8}$$

where $action_{pre}$ is the action generated in the previous step of CCRRL, and modify represents the adjustment of the range of values for the rewards.

In the fourth strategy, denoted as $S4$, $action_{adv}$ is generated by multiplying $action_{pre}$ and the absolute value of $reward$. The mathematical representation of $action_{adv}$ is given by Equation 9:

$$(\textbf{S4}) \quad action_{adv} = action_{pre} \times |reward| \tag{9}$$

where $action_{pre}$ is the action generated in the previous step of CCRRL.

### 4.4 AGENT'S FINAL ACTION

The ultimate action produced by the CCRRL model in its current state is denoted as $action_{fin}$. This action is determined by considering both the opponent's action and the CCRRL's previous actions. Specifically, we combine $action_t$ with $action_{adv}$ to obtain $action_{fin}$ as the final action for the agent. This equation incorporates the adversarial action $action_{adv}$ into the computation of the agent's ultimate action. The mathematical representation of $action_{adv}$ is provided in Equation 10:

$$action_{fin} = action_t + action_{adv} \tag{10}$$

where $action_t$ is the action generated by an agent at time step $t$ according to the state $s_t$, and $action_{adv}$ represents the action of an adversary agent.

## 4.5 MODIFICATION FUNCTION

The concept of $modify$ enables the adjustment of reward value ranges during the generation step of $action_{adv}$. This adjustment facilitates fine-tuning of reward values, contributing to improved decision-making and performance optimization. The modify function takes a reward value denoted by $reward$, which is generated in the previous step, as input and performs the following operations:

- If the reward value is 0, it returns 0.

- If the absolute value of the reward is less than 1, the reward value is repeatedly multiplied by 10 until its absolute value becomes greater than 1.

- If the absolute value of the reward is greater than 1, it remains unchanged.

Therefore, the modify function ensures that the reward value is adjusted in such a way that the absolute value of the non-zero input is greater than or equal to 1, while returning zero for a zero input. This adjustment process allows for a more effective scaling and normalization of reward values within the CCRRL framework.

## 5 CCRRL FOR THE PORTFOLIO OPTIMIZATION

This section aims to describe the key steps of CCRRL for generating portfolio strategies. The CCRRL model proposed in this study comprises four networks: the actor network $\mu(s|\theta_\mu)$, critic network $Q(s, action|\theta_Q)$, target actor network $\mu'(s|\theta_{\mu'})$, and target critic network $Q'(s, action|\theta_{Q'})$. The algorithmic steps for CCRRL are outlined in Algorithm 1, providing a detailed procedure for the implementation and execution of CCRRL.

---

**Algorithm 1** CCRRL for portfolio selection.

---

1: Initialize the actor network, critic network, target actor network, and target critic network with random weights.
2: Set the replay buffer $D$ to an empty memory.
3: Set the exploration noise process.
4: **while** The termination condition is not satisfied **do**
5:     Initialize the portfolio with an initial allocation.
6:     Reset the environment and obtain the initial state $s_0$.
7:     Set the total episode reward $R$ to 0.
8:     **for** each period in the episode **do**
9:         Random select a strategy $S_{fixed}$ from S1, S2, S3 and S4 (Section 4.2.1).
10:        **for** each time step $t$ in the period **do**
11:            Select $action_t$ using the current policy and exploration noise (Section 4.1).
12:            Random generate adversary action $action_{adv}$ using $S_{fixed}$.
13:            Combine $action_t$ with $action_{adv}$ to produce $action_{fin}$ (Section 4.4).
14:            Execute $action_{fin}$ and observe $reward_t$ and the new state $s_{t+1}$.
15:            Store the transition $(s_t, action_t, reward_t, s_{t+1})$ in the replay buffer $D$.
16:            Sample a random transitions $(s_i, action_i, reward_i, s_{i+1})$ from the replay buffer $D$.
17:            Update the critic network parameters by minimizing the loss function.
18:            Update the actor network parameters using the sampled policy gradient.
19:            Update the target networks.
20:            Set $s_t = s_{t+1}$ and $R = R + reward_t$.
21:        **end for**
22:     **end for**
23: **end while**
24: Return the portfolio strategy with the best total reward found thus far.

---

CCRRL commences by initializing the actor network, critic network, target actor network, and target critic network with randomly assigned weights (step 1). At step 2, an empty memory, referred to as the replay buffer $D$, is instantiated, serving as a repository for experience replay. At step 3, an exploration noise process is established to facilitate exploration during the learning process.

The main loop (step 4- 23) stops until a specified termination condition is met. Within each iteration, the portfolio is initialized with an initial allocation (step 5). The environment is reset, and the initial

state $s_0$ is obtained (step 6). A running tally of the total episode reward, denoted as $R$, is set to zero (step 7).

For each time step $t$ within the episode, the algorithm proceeds as follows. The current policy $\mu(s_t|\theta_\mu)$, incorporating exploration noise $\mathcal{N}$, is utilized to select an action $action_t$ (step 11). At step 12, an action $action_{adv}$ is randomly generated among S1, S2, S3 and S4 (detailed in Section 4.2.1). The actions $action_t$ and $action_{adv}$ are then combined to produce a final action $action_{fin}$ (step 13). At step 14, the chosen action $action_{fin}$ is executed, resulting in the observation of a reward $reward_t$ and the subsequent state $s_{t+1}$. Then, the transition $(s_t, action_t, reward_t, s_{t+1})$ is stored in the replay buffer $D$ (step 15). At step 16, a random minibatch of transitions $(s_i, action_i, reward_i, s_{i+1})$ is sampled from the replay buffer $D$.

At step 17, the critic network parameters are updated by minimizing the loss function:

$$L = \frac{1}{N} \sum_i (y_i - Q(s_i, action_i|\theta_Q))^2,$$

where $y_i = reward_i + \gamma Q'(s_{i+1}, \mu'(s_{i+1}|\theta_{\mu'})|\theta_{Q'})$ and $\gamma$ is the discount factor. At step 18, the actor network parameters are updated using the sampled policy gradient:

$$\nabla_{\theta_\mu} J \approx \frac{1}{N} \sum_i \nabla_{action} Q(s, action|\theta_Q)|_{s=s_i, action=\mu(s_i)} \nabla_{\theta_\mu} \mu(s|\theta_\mu)|_{s_i}$$

At step 19, the target networks are also updated to enhance stability during training:

$$\theta_{\mu'} \leftarrow \tau\theta_\mu + (1-\tau)\theta_{\mu'}, \quad \theta_{Q'} \leftarrow \tau\theta_Q + (1-\tau)\theta_{Q'}$$

Furthermore, the current state $s_t$ is updated to $s_{t+1}$, and the episode reward $R$ is incremented by $reward_t$ (step 20). Once the termination condition is met, the algorithm concludes by returning the portfolio strategy yielding the highest total reward discovered thus far (step 24).

## 6 EMPIRICAL ANALYSIS

The CCRRL model proposed in this research was developed using Python 3.8, utilizing the state-of-the-art FIN-RL library introduced by Liu et al. Liu et al. (2022). Our experiments were conducted on a system with an Intel i7-7700 CPU, running the Microsoft Windows 10 operating system. The system was equipped with 16 GBytes of RAM and a Nvidia GTX 1050 Ti GPU. To ensure the reliability and statistical significance of our results, we performed 100 random independent trials for each game strategy. This extensive experimentation allows for a robust evaluation of the performance of the CCRRL model and the comparison of different strategies. In addition, we compared the performance of the CCRRL model with state-of-the-art methods, including A2C (Advantage Actor-Critic) Mnih et al. (2016), PPO (Proximal Policy Optimization) Schulman et al. (2017), and the traditional DDPG (Deep Deterministic Policy Gradient) Lillicrap et al. (2015). This comparison enables us to assess the effectiveness and superiority of the proposed CCRRL approach against existing state-of-the-art techniques.

### 6.1 DATA DESCRIPTION

In this research, we focused on the ChiNext market, which is characterized by strict regulations, high supervision, elevated risk levels, and relatively lower entry thresholds for listed companies. To conduct our analysis, we obtained daily stock trading data for all the stocks listed on the ChiNext market from the reputable China Stock Market and Accounting Research (CSMAR) database. We performed thorough data cleaning procedures to ensure the quality and reliability of the dataset. The resulting data sample consists of 240 ChiNext stocks traded from January 1, 2018, to December 31, 2020, covering a span of 730 trading days. To provide a comprehensive understanding of the technical indicators used in the RRL-MAS model, we refer readers to Table 3 in the Appendix, which presents a detailed overview of the indicators incorporated in our study.

## 6.2 PERFORMANCE OF CCRRL

In order to assess the robustness of the CCRRL model, we have employed a comprehensive set of risk measurement indicators. These indicators play a crucial role in evaluating the performance and reliability of portfolio strategies provided by CCRRL under different scenarios. For a detailed explanation of each indicator, please refer to Table 4 in the Appendix. To further analyze the effectiveness of the CCRRL model, we conducted a series of experiments using two types of data: linear and non-linear structured data. Linear structured data pertains to information solely focusing on the stock trading technical indicators. On the other hand, non-linear structured data incorporates investor attention as an influential factor in the analysis. This distinction enables us to understand the advantages of models when dealing with different types of data.

Figure 2 illustrates the comparison of daily cumulative returns among the A2C, PPO, DDPG and CCRRL (Algorithm 24) for a single run. From Figure 2, we can observe that the curves of A2C and PPO predominantly remain below both DDPG and CCRRL. This observation indicates that CCRRL consistently exhibit superior performance in terms of daily cumulative returns. Moreover, Table 5 displays the detailed results provided by the proposed game strategies (see Appendix). While the detailed results provided by CCRRL with different period size are shown in Table 6. We can remark that, when the period size is equal to 15, the results provide by CCRRL are most stable.

| Indicators | A2C $Mean$ | $Std$ | PPO $Mean$ | $Std$ | DDPG $Mean$ | $Std$ | CCRRL (Step=15) $Mean$ | $Std$ |
|---|---|---|---|---|---|---|---|---|
| Annual Return | 0.193 | 0.245 | 0.082 | 0.128 | 0.247 | 0.066 | 0.265 | 0.086 |
| Cumulative Return | 0.229 | 0.292 | 0.096 | 0.150 | 0.289 | 0.078 | 0.311 | 0.103 |
| Annual Volatility | 0.377 | 0.070 | 0.323 | 0.014 | 0.308 | 0.010 | 0.311 | 0.017 |
| Sharpe Ratio | 0.590 | 0.511 | 0.392 | 0.363 | 0.869 | 0.174 | 0.907 | 0.215 |
| Calmar Ratio | 0.772 | 0.960 | 0.434 | 0.663 | 1.246 | 0.372 | 1.341 | 0.479 |
| Stability | 0.455 | 0.283 | 0.398 | 0.245 | 0.736 | 0.062 | 0.739 | 0.070 |
| Max Drawdown | -0.284 | 0.082 | -0.239 | 0.044 | -0.201 | 0.016 | -0.201 | 0.024 |
| Omega Ratio | 1.114 | 0.102 | 1.071 | 0.066 | 1.158 | 0.033 | 1.166 | 0.042 |
| Sortino Ratio | 0.827 | 0.727 | 0.527 | 0.493 | 1.187 | 0.245 | 1.236 | 0.303 |
| Tail Ratio | 0.996 | 0.165 | 0.883 | 0.071 | 0.942 | 0.065 | 0.946 | 0.061 |
| Value at Risk | -0.047 | 0.008 | -0.040 | 0.002 | -0.038 | 0.001 | -0.038 | 0.002 |
| Alpha | -0.235 | 0.233 | -0.340 | 0.113 | -0.269 | 0.074 | -0.246 | 0.088 |
| Beta | 0.933 | 0.066 | 0.927 | 0.037 | 0.956 | 0.036 | 0.956 | 0.042 |

Table 2: Comparative study among the A2C, PPO, DDPG and CCRRL.

Table 2 presents a comparative study among the A2C, PPO, DDPG and CCRRL. These strategies are evaluated based on various financial performance indicators, and the table provides both the mean and standard deviation (Std) values for each indicator. The experimental findings indicate that CCRRL appears to outperform the other strategies in terms of key performance indicators, including annual return, Sharpe ratio, Calmar ratio, Omega ratio.

## 7 CONCLUSION

This study presents an intelligent portfolio optimization model based on reinforcement learning techniques. By conceptualizing the portfolio optimization problem as a multi-agent game, we introduce multiple autonomous agents that actively engage in competition and cooperation within a realistic market environment. Applying a deep reinforcement learning framework, these agents are trained to continually learn and adapt their investment strategies over time. To empirically evaluate the efficacy of our proposed approach, we conduct extensive experiments using real historical stock trading data. The results substantiate the superiority of our competitive and cooperative strategies robust reinforcement learning framework over traditional reinforcement learning approaches.

## ACKNOWLEDGMENTS

We would like to thank the anonymous reviewers for their relevant and rich remarks that allowed us to improve the presentation of our results.

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

# A APPENDIX

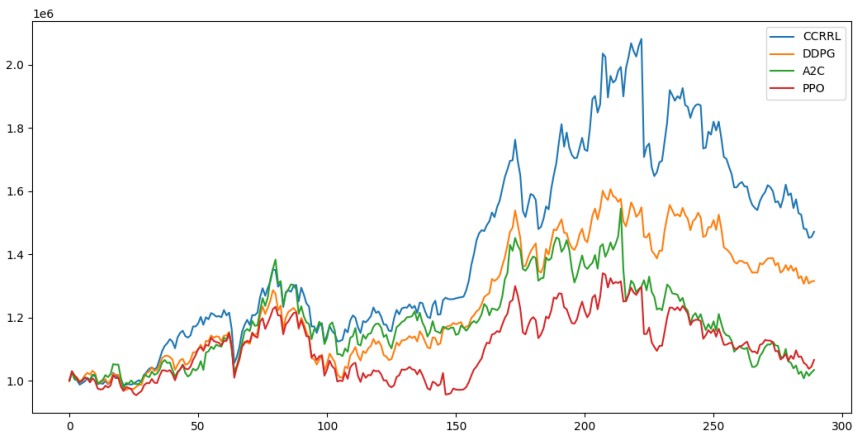

Figure 2: Comparative study among the A2C, PPO, DDPG and CCRRL.

| Indicators | Explanation |
|---|---|
| Close | The closing price of the day's trading. |
| Open | The opening price of the day's trading. |
| Hight | The highest point that a stock can reach during a day's trading. |
| Low | The lowest point that a stock can reach during a day's trading. |
| Volume | The total amount of trading activity in a given financial market. |
| Turnover | The total number of shares traded during a certain period of time. |
| Boll_ub | The upper band of the Bollinger Bands indicator. |
| Boll_lb | The lower band of the Bollinger Bands indicator. |
| SMA_30 | The simple moving average of a stock's closing price over the past 30 days. |
| SMA_60 | The simple moving average of a stock's closing price over the past 60 days. |
| MACD | Moving Average Convergence Divergence, which shows the relationship between two exponential moving averages and is used to identify potential trend changes. |
| RSI_30 | The Relative Strength Index (RSI) within 30 time periods, where RSI measures the magnitude of recent price changes to evaluate overbought or oversold conditions in the price of a stock or other asset. |
| CCI_30 | The product channel index over a period of 30 time periods, where CCI (Commodity Channel Index) is an indicator that examines the relationship between a fluctuating asset's price, moving average, and the normal deviations from that average. |
| DX_30 | The directional movement index over a period of 30 time periods, which measures the strength and direction of a trend by comparing the current price to a previous price and displays positive or negative directional movement. |

Table 3: Description of the used stock technical indicators.

| Indicators | Explanation |
|---|---|
| Annual Return | It is the average rate of return an investment generates each year over a specified period of time. |
| Cumulative Return | It refers to the total amount of return earned or lost on an investment over a certain period of time. |
| Annual Volatility | It is a measure of the amount of variation or fluctuation in the returns of an investment over a one-year period. |
| Sharpe Ratio | It is used to measure the relationship between yield and risk-free interest rate. |
| Calmar Ratio | The formula for the Calmar ratio is "excess return / max drawdown", which directly measures the relationship between return and max drawdown. |
| Stability | It examines the historical volatility in stock prices and evaluates the consistency of income generated to ascertain its stability. |
| Max Drawdown | It describes the worst-case scenario that a strategy can have, which is the extent to which net worth of a product begins to fall from its highest point to its lowest point over a period of time. |
| Omega Ratio | It is a risk-adjusted performance measure used to evaluate an investment's return in relation to the risk of negative returns. |
| Sortino Ratio | It is a risk-adjusted performance measure used to evaluate the return of an investment relative to its downside risk. |
| Tail Ratio | It is a financial risk metric used to evaluate an investment's potential risk and reward by comparing the size of its left tail to that of its right tail. |
| Value at Risk | It is a statistical measure used to estimate the potential loss on an investment. |
| Alpha | It refers to the risk associated with the specific investments within a portfolio that cannot be attributed to general market movements. |
| Beta | It represents the portion of investment risk that is attributable to overall market movements. |

Table 4: Indicators used to evaluate portfolio strategies.

| | S1 | | | | S2 | | | | S3 | | | | S4 | | | |
|---|---|---|---|---|---|---|---|---|---|---|---|---|---|---|---|---|
| Indicators | Min | Max | Mean | Std | Min | Max | Mean | Std | Min | Max | Mean | Std | Min | Max | Mean | Std |
| Annual Return | 0.079 | 0.518 | 0.275 | 0.084 | 0.018 | 0.446 | 0.254 | 0.087 | 0.057 | 0.411 | 0.250 | 0.076 | 0.033 | 0.441 | 0.233 | 0.074 |
| Cumulative Return | 0.091 | 0.616 | 0.323 | 0.100 | 0.021 | 0.529 | 0.298 | 0.104 | 0.066 | 0.487 | 0.293 | 0.091 | 0.038 | 0.522 | 0.273 | 0.088 |
| Annual Volatility | 0.283 | 0.351 | 0.310 | 0.012 | 0.285 | 0.356 | 0.308 | 0.011 | 0.284 | 0.330 | 0.307 | 0.010 | 0.282 | 0.340 | 0.308 | 0.010 |
| Sharpe Ratio | 0.398 | 1.497 | 0.934 | 0.206 | 0.213 | 1.352 | 0.885 | 0.229 | 0.336 | 1.282 | 0.878 | 0.205 | 0.259 | 1.354 | 0.834 | 0.198 |
| Calmar Ratio | 0.355 | 2.563 | 1.400 | 0.439 | 0.075 | 2.406 | 1.287 | 0.483 | 0.235 | 2.274 | 1.280 | 0.451 | 0.127 | 2.304 | 1.186 | 0.421 |
| Stability | 0.540 | 0.871 | 0.753 | 0.062 | 0.201 | 0.858 | 0.733 | 0.089 | 0.435 | 0.848 | 0.736 | 0.074 | 0.336 | 0.847 | 0.721 | 0.077 |
| Max Drawdown | -0.252 | -0.152 | -0.198 | 0.018 | -0.278 | -0.159 | -0.201 | 0.020 | -0.252 | -0.149 | -0.200 | 0.019 | -0.260 | -0.161 | -0.200 | 0.019 |
| Omega Ratio | 1.070 | 1.283 | 1.170 | 0.040 | 1.037 | 1.251 | 1.161 | 0.044 | 1.058 | 1.242 | 1.159 | 0.039 | 1.045 | 1.252 | 1.151 | 0.038 |
| Sortino Ratio | 0.535 | 2.078 | 1.278 | 0.289 | 0.283 | 1.883 | 1.213 | 0.322 | 0.458 | 1.777 | 1.201 | 0.286 | 0.342 | 1.863 | 1.139 | 0.277 |
| Tail Ratio | 0.826 | 1.068 | 0.935 | 0.055 | 0.808 | 1.083 | 0.940 | 0.061 | 0.808 | 1.125 | 0.937 | 0.062 | 0.770 | 1.107 | 0.945 | 0.066 |
| Value at Risk | -0.043 | -0.035 | -0.038 | 0.001 | -0.044 | -0.035 | -0.038 | 0.001 | -0.041 | -0.035 | -0.038 | 0.001 | -0.042 | -0.035 | -0.038 | 0.001 |
| Alpha | -0.447 | -0.007 | -0.243 | 0.081 | -0.446 | -0.029 | -0.253 | 0.085 | -0.438 | -0.056 | -0.264 | 0.087 | -0.461 | -0.067 | -0.282 | 0.078 |
| Beta | 0.860 | 1.053 | 0.966 | 0.041 | 0.860 | 1.039 | 0.961 | 0.039 | 0.869 | 1.047 | 0.951 | 0.038 | 0.881 | 1.061 | 0.949 | 0.035 |

Table 5: Comparative study between Robust Strategies: S1, S2, S3 and S4.

| | Step=1 | | | | Step=5 | | | | Step=10 | | | | Step=15 | | | |
|---|---|---|---|---|---|---|---|---|---|---|---|---|---|---|---|---|
| Indicators | Min | Max | Mean | Std | Min | Max | Mean | Std | Min | Max | Mean | Std | Min | Max | Mean | Std |
| Annual Return | 0.078 | 0.417 | 0.251 | 0.071 | 0.087 | 0.390 | 0.257 | 0.064 | 0.101 | 0.651 | 0.258 | 0.097 | 0.107 | 0.477 | 0.265 | 0.086 |
| Cumulative Return | 0.091 | 0.493 | 0.295 | 0.085 | 0.101 | 0.461 | 0.301 | 0.077 | 0.117 | 0.781 | 0.303 | 0.116 | 0.124 | 0.567 | 0.311 | 0.103 |
| Annual Volatility | 0.283 | 0.333 | 0.305 | 0.010 | 0.277 | 0.376 | 0.309 | 0.014 | 0.275 | 0.367 | 0.309 | 0.016 | 0.281 | 0.383 | 0.311 | 0.017 |
| Sharpe Ratio | 0.405 | 1.323 | 0.887 | 0.192 | 0.426 | 1.256 | 0.896 | 0.169 | 0.457 | 1.638 | 0.892 | 0.240 | 0.496 | 1.460 | 0.907 | 0.215 |
| Calmar Ratio | 0.327 | 2.378 | 1.297 | 0.406 | 0.386 | 2.225 | 1.330 | 0.383 | 0.450 | 3.382 | 1.306 | 0.531 | 0.524 | 2.463 | 1.341 | 0.479 |
| Stability | 0.449 | 0.855 | 0.738 | 0.074 | 0.552 | 0.852 | 0.747 | 0.050 | 0.508 | 0.870 | 0.736 | 0.077 | 0.458 | 0.852 | 0.739 | 0.070 |
| Max Drawdown | -0.240 | -0.165 | -0.196 | 0.016 | -0.320 | -0.155 | -0.197 | 0.021 | -0.310 | -0.160 | -0.201 | 0.021 | -0.327 | -0.162 | -0.201 | 0.024 |
| Omega Ratio | 1.071 | 1.246 | 1.161 | 0.036 | 1.073 | 1.235 | 1.163 | 0.032 | 1.083 | 1.328 | 1.163 | 0.047 | 1.087 | 1.274 | 1.166 | 0.042 |
| Sortino Ratio | 0.545 | 1.835 | 1.212 | 0.270 | 0.582 | 1.733 | 1.224 | 0.235 | 0.596 | 2.244 | 1.215 | 0.336 | 0.668 | 2.014 | 1.236 | 0.303 |
| Tail Ratio | 0.821 | 1.111 | 0.952 | 0.053 | 0.824 | 1.082 | 0.944 | 0.057 | 0.809 | 1.099 | 0.946 | 0.059 | 0.797 | 1.186 | 0.946 | 0.061 |
| Value at Risk | -0.041 | -0.034 | -0.037 | 0.001 | -0.046 | -0.034 | -0.038 | 0.002 | -0.045 | -0.033 | -0.038 | 0.002 | -0.047 | -0.035 | -0.038 | 0.002 |
| Alpha | -0.406 | -0.019 | -0.254 | 0.069 | -0.427 | -0.052 | -0.257 | 0.069 | -0.431 | 0.047 | -0.248 | 0.087 | -0.439 | -0.026 | -0.246 | 0.088 |
| Beta | 0.854 | 1.025 | 0.950 | 0.036 | 0.859 | 1.059 | 0.959 | 0.037 | 0.847 | 1.049 | 0.950 | 0.043 | 0.859 | 1.120 | 0.956 | 0.042 |

Table 6: Performance of CCRRL with different parameter settings.

