# OpenReview forum: "Robust Reinforcement Learning for Portfolio Management via Competition and Cooperation Strategies"
_ICLR.cc/2024/Conference — ICLR 2024 Conference Withdrawn Submission_

### Official Review · Reviewer_Lxrc · 2023-10-25

**Soundness:** 2 fair
**Presentation:** 3 good
**Contribution:** 2 fair
**Rating:** 5
**Confidence:** 2

**Summary:**

This work develop a framework for portfolio management using robust RL. An adversary is introduced to model the uncertainty in the real application, and the goal is to find a policy performs well under the uncertainty.

**Strengths:**

1. The idea to apply robust RL in this area is interesting, considering the inherent uncertainty in the problem. The idea of using an adversarial is not new, but still interesting and useful.
2. According to the experiment results, the CCRRL approach outperforms other RL algorithms, which shows the power of the framework.

**Weaknesses:**

I am not an expert in finance, but from the aspect of machine learning or reinforcement learning, the contribution of this paper is not much. The idea of adversarial training has been used for a while, and I can't see too much about this paper's contribution or novelty.

I somehow feel that this paper does not fit in the ICLR conference, considering the major contribution and motivation. I admit that I am not familiar with finance area, so I maintain a low confidence score.

**Questions:**

I don't have any questions.

---

> ### Author Response · Authors · 2023-11-23
>
> In the new version, we have made improvements to the literature review and model mechanism explanation sections, and changes have been marked in red in the main text.

---

### Official Review · Reviewer_Nhcj · 2023-10-25

**Soundness:** 2 fair
**Presentation:** 2 fair
**Contribution:** 2 fair
**Rating:** 3
**Confidence:** 3

**Summary:**

This paper proposes a robust reinforcement learning (RL) approach within a multi-agent framework to portfolio optimization. This approach considers both cooperation and competition strategies: Competition lets the agents compete for limited resources and cooperation is used to accelerate convergence. Four different cooperation/competition strategies are proposed where the difference comes from the different modifications applied to the selected actions before they are executed. The authors conducted experiments on a real-world financial dataset that they collected, and they made comparisons with standard single-agent RL algorithms including A2C, PPO, and DDPG.

**Strengths:**

This paper proposes to study the portfolio optimization problem from an interesting perspective, by formulating this problem as a multi-agent system with both cooperation and competition.

**Weaknesses:**

In my opinion, the amount of contribution and the significance of the results in the paper are not sufficient for ICLR. The writing/presentation is vague and non-rigorous throughout the paper, which prevents me from further appreciating the contributions of the work.

Specifically, this paper proposes a new multi-agent formulation, but fails to properly define many important concepts in their approach, such as the definition of the state and action spaces. The definitions of the game formulation (e.g., who the players are and how they are cooperative/competitive) and the state transitions are also missing. It is unclear to me how the RL actions translate to the decisions in the portfolio optimization problem. The four proposed strategies (by multiplying the previously selected action to a certain value) are also a bit arbitrary to me and there is no justification about why such strategies are chosen. The authors also do not discuss how the reward functions are defined and why they need a further modification function to the rewards.

For the methodology part, the authors simply use an existing and standard DDPG algorithm for learning. In the experiments, the authors only evaluated their method on one dataset that they collected, without introducing the formats or the basic statistics of the dataset. The authors also only compared their method with the performances of standard single-agent RL algorithms despite that their formulation is multi-agent, which makes me question if such comparisons are fair. These experiment results are not sufficient to make conclusions about the effectiveness of the proposed method.

**Questions:**

How do you define the action space in the RL formulation, and how do they translate to the decisions in your portfolio optimization problem?

Your CCRRL method uses DDPG as the RL solver, but why would you compare it with another DDPG algorithm in your experiments? How are these DDPG solvers different from each other?

---

> ### Author Response · Authors · 2023-11-23
>
> In the new version, we have made improvements to the literature review and model mechanism explanation sections, and changes have been marked in red in the main text.
>
> The action space is [-1,1], and each model will generate an action array with a length of the number of stocks. Each element represents the number of stocks sold or bought for this stock, and the positive sign represents the purchase and the negative sign represents the sale. In the actual processing, it will be multiplied by 1000 to restore the true value range. The DDPG model used by CCRRL is a modified DDPG with the addition of S1-S4 methods. Compared with the original DDPG method, it highlights the improvement of the results after adding S1-S4. A2C, PPO, and DDPG are three single agents, and it is not appropriate to compare them with multi-agent.

---

### Official Review · Reviewer_Ju8B · 2023-10-27

**Soundness:** 1 poor
**Presentation:** 1 poor
**Contribution:** 1 poor
**Rating:** 3
**Confidence:** 4

**Summary:**

The paper proposes an approach to robust portfolio optimization via reinforcement learning (RL). Since RL is known to be sensitive to dynamic changes in the environment, the paper proposes CCRRL (Competitive and Cooperative Robust Reinforcement Learning via Multi-Agent System), which combines techniques from cooperative and competitive MARL. The approach is based on DDPG and is evaluated in some financial markets setting.

**Strengths:**

The paper addresses an interesting application domain.

**Weaknesses:**

**Novelty**

The paper applies DDPG to a specific domain. There are no novel additions to the algorithm itself. Instead, the original policy actions are modified according to some rules to add cooperative and competitive aspects (I do not understand how and why, though). Overall, I consider the novelty of the contributions as limited.

**Clarity**

I do not understand the motivation behind this approach. To me, the whole market setting is selfish (neither purely cooperative nor competitive) thus it is unclear to me how the cooperative and competitive additions are supposed to help in this case. Unfortunately, the paper does not provide any theoretical reasoning but only keeps the concepts on a high level by explaining how things are done - but not why.

Despite introducing variables in Table 1, there is no formal definition of the RL setting and how it connects with the portfolio optimization problem (e.g., how do state and action space look like? What is the actual reward structure of the problem?).

I also do not understand why modified rewards are multiplied to actions. What is rationale behind multiplying actions and rewards?

Furthermore, I do not see any multi-agent component in the whole approach. The market might represent a selfish multi-agent system, but as far as I understood, only single-agent RL is applied to a particular agent. There are no multi-agent-specific techniques used in CCRRL, such as credit-assignment mechanisms, counterfactual baselines, centralized critics, etc.

**Soundness**

The competitive and cooperative part are not sound in my view.

The competitive part only multiplies the action with a modified reward (that preserves the sign) but does not take the rewards of others into account (in competitive settings, the goal is to minimize other agents' rewards, which is impossible if they are not known). Therefore, a "competitive" agent would merely be an individual rational player who acts selfishly in a game-theoretic sense.

The cooperative part only considers the absolute value of the reward, which could be negative. In a true cooperative setting, all agents perceive a shared reward and optimize a common value function. In the market setting, the agent therefore needs to consider the reward of all other agents, which is unrealistic for the financial market domain.

Having some kind of theoretical analysis that could support the assumptions and approaches would be appreciated. I further doubt the soundness of the cooperative part due to the following example:
- Given a Prisoner's dilemma with a cooperation reward of $R = -1$, a defection reward of $P = -2$, a sucker’s reward of $S = -3$, and a temptation reward of $T = 0$.
- According to the paper's definition, a cooperative agent would aim for $S = -3$ due to the reward having the highest absolute value.
- In this case, a "cooperative" player would allow itself to be exploited by the other player.
- However, this would neither be optimal for the cooperative player itself (it would actually be the individual worst case with a reward of $S = -3$) nor for the whole system (the true optimum is mutual cooperation with a common reward of $R = -1$ per player).

Thus, I suppose that the method does not work for general self-interested games. Further analysis of suitable games would be helpful to assess the contribution of the paper.

**Significance**

I do not consider the concepts and results as particularly significant. The paper merely compares with 3 standard RL algorithms (A2C, PPO, DDPG). No multi-agent algorithm is used for comparison.

**Minor Comments**
- It would help in Section 4.2 to explicitly refer to Figure 1, otherwise a reader may not be aware of the meaning of S1, S2, etc. at this point.
- The modified reward function can be defined with a simple equation as follows: $max(r, ceil(r) - floor(r))$

**Questions:**

1. What does MAG-RL stand for in Figure 1?
2. *“This opponent is generated by the CCRRL model based on the output of the previous step.”* - How is this done? Does the opponent use the same model?

---

> ### Author Response · Authors · 2023-11-23
>
> In the new version, we have made improvements to the literature review and model mechanism explanation sections, and changes have been marked in red in the main text.
>
> Opponent is created using S1-S4, and its significance lies in using the results of the previous step to guide the direction of action generation in this step.

---

### Official Review · Reviewer_vSqN · 2023-10-28

**Soundness:** 1 poor
**Presentation:** 2 fair
**Contribution:** 1 poor
**Rating:** 3
**Confidence:** 3

**Summary:**

This paper proposes multi-agent RL framework for robust decision-making for portfolio optimization in financial institutions.  The paper uses a DDPG framework to take action and then consider how an adversary or cooperative multi-agent RL framework can address the challenges.

**Strengths:**

The paper seeks to use multi-agent framework to take robust action. Finding robust action is important in MDP setup. The algorithm seems to outperform the baseline algorithms in terms of reducing the risk.

**Weaknesses:**

1. The paper is not well-written as a lot of ideas have not been well-explained. For example, what is the reward? Why one has to use a modification function? Why is the adversarial action taken like in S1 and S2? Why is the cooperative action taken like in S3, and S4?

2. How each of the S1, S2, S3, and S4 are appended with the action?

3. The paper does not provide any theoretical result.

**Questions:**

Please see the weakness.

---

> ### Author Response · Authors · 2023-11-23
>
> In the new version, we have made improvements to the literature review and model mechanism explanation sections, and changes have been marked in red in the main text.
>
> In the model, the states include the current total funds, the closing prices of each stock, and the current holdings of each stock.
>
> The reward calculation first determines the total funds before and after the market opens and closes, takes the difference between them, and then subtracts the transaction fees to obtain the reward for the day. Finally, the reward value is multiplied by 0.001 to narrow its range.
>
> In the model, after the action is generated, we apply strategies S1-S4 to process the action.

---

### Official Review · Reviewer_AQy4 · 2023-10-30

**Soundness:** 2 fair
**Presentation:** 2 fair
**Contribution:** 1 poor
**Rating:** 3
**Confidence:** 2

**Summary:**

The paper extends current DDPG method for Portfolio Optimization method with a modified action space. The aim for processing the actions is for more robust actor performance.

**Strengths:**

The paper is written with good clarity on the domain and related works.

**Weaknesses:**

The definition of the task/environment is vague. The proposed action generation process is not considered part of the intrinsics to the agent, but not part of the environment as well. Therefore, the problem itself is no longer a RL problem, a further definitive narrative is needed to justify usage of RL methods. It is also confusing, since the manuscript mentions that DDPG generates reward, but in fact I do not think that is the case. The action is also not defined clearly.
The authors mention that this is a work on multi-agent learning, or that the problem is a multi-agent environment. However, I fail to see how this formulation manifest in the problem or in the proposed method. I view the action generation process an additional function on top of learned actions, which is not theoretically justified in any way.

**Questions:**

How would you argue the correctness of the method? How would you show its convergence in theory?
What is the motivation for the action modifications?

---

> ### Author Response · Authors · 2023-11-23
>
> In the new version, we have made improvements to the literature review and model mechanism explanation sections, and changes have been marked in red in the main text.
>
> The motivation of the modify function is to consider the reward of the previous action and adjust the action of this step accordingly. If the reward of the previous step is negative, it indicates that the previous action was not good and needs to be avoided in order to achieve robustness and improve reward.

---

### Official Review · Reviewer_3VrD · 2023-10-30

**Soundness:** 2 fair
**Presentation:** 1 poor
**Contribution:** 2 fair
**Rating:** 3
**Confidence:** 4

**Summary:**

This paper proposes a robust RL algorithm for portfolio optimization. The authors adopt DDPG with action perturbation by four kinds of cooperation or competitive strategies. The consequent method can have better performance than regular RL algorithms like PPO, A2C, and DDPG.

**Strengths:**

1. The authors provide a method to enhance performance of RL policies for portfolio optimization.
2. The authors propose four different strategies for the robust purpose of non-stationary environments.

**Weaknesses:**

1. A majority of this paper is not well-written. The authors should revise both the format and the content of this paper.
2. The proposed method seems not novel and does not show a superior performance compared to basic RL baselines.
3. The experiments may be insufficient to show the method's effectiveness and robustness.

Some minor errors in the paper:
1. Formula in Figure 1 are blurry.
2. Notations like $reward$ to represent the reward in previous step is wired.

**Questions:**

1. How does the mathematical program correlate with the RL part? For instance, how can a DDPG algorithm optimize the problem in Equation (1)-(4), including the constraints?
2. In Equation (3), why do we include the constraint on the expected return, given the fact that it should be maximized?
3. In Equation (4), how can a RL algorithm constrain the variance?
4. What is the form of the `modify` function?
5. How does the proposed method, Robust Reinforcement Learning via Multi-Agent System (CCRRL), correlate with the multi-agent system? The developed strategies are not agent-level but permutation along the action domain.

---

> ### Author Response · Authors · 2023-11-23
>
> In the new version, we have made improvements to the literature review and model mechanism explanation sections, and changes have been marked in red in the main text.
>
> In our model, we avoid the previous step's reward having too much influence on the current step's action. Because the result of the previous step only has a reference effect, it is not possible to copy it completely.
>
> The subjects of cooperation and competition are the four models trained with s1-s4. There is no data interaction between them, except that in each round, the model with the best reward in the simulation is selected to provide actions for training ddpg. However, they share the same environment and there is indirect data interaction. The system in this case can be regarded as a distributed agent system.
>
> The function of Modify can be expressed as:
> \begin{equation}
> 	modify(reward) = \begin{cases}
> 		0 & \text{if } reward = 0 \\
> 		10^n \times reward & \text{if } |reward| < 1\\
> 		reward & \text{if } |reward|>1
> 	\end{cases}
> 	\label{equation2}
> \end{equation}

---

### Official Review · Reviewer_T3xy · 2023-10-31

**Soundness:** 2 fair
**Presentation:** 2 fair
**Contribution:** 2 fair
**Rating:** 3
**Confidence:** 3

**Summary:**

This paper proposes a portfolio optimization model based on reinforcement learning and introduces competition and cooperation strategies via multi-agent games to enhance the robustness of the model.

**Strengths:**

The proposed method incorporates both competition and cooperation strategies to enhance decision-making performance and adaptability. By formulating the portfolio management problem as a multi-agent environment, agents collaborate and jointly strive to achieve a common goal.

**Weaknesses:**

This paper in fact proposes a multi-agent reinforcement learning method for portfolio optimization problem. However, no MARL methods are discussed and compared with the proposed method.  Further, the proposed method only makes modifications on actions of agents based on rewards, which has no novelty.

**Questions:**

1. The author claims that they introduce both competition and cooperation strategies in the proposed method. However, this will lead to unstationary learning environment, which makes the method hard to be converged. In fact, most MARL methods work only for cooperative cases.
2. In the experiment, no MARL methods are used as baselines.

---

> ### Author Response · Authors · 2023-11-23
>
> In the new version, we have made improvements to the literature review and model mechanism explanation sections, and changes have been marked in red in the main text.
>
> Comment 1: Because the four models do not affect the environment simultaneously, they can solve the problem of unstable environments. However, it is unclear whether multiple agents that do not affect the environment simultaneously are truly multi-agent environments.
> Comment 2: We conducted a search and found no open-source projects in the current literature that use MARL to solve dynamic portfolio optimization problems.

---

### Official Review · Reviewer_uHcE · 2023-10-31

**Soundness:** 2 fair
**Presentation:** 2 fair
**Contribution:** 2 fair
**Rating:** 3
**Confidence:** 4

**Summary:**

The authors propose a multi-agent reinforcement learning approach for portfolio optimization. The authors call their MARL approach "robust", and claim that it uses both cooperation and collaboration. The algorithm appears to build on top of DDPG.

The claims of the paper include that the system can "mitigate the negative impact of the variations of the market" and that it leads to a "collective enhancements in the performance of RL algorithms".

**Strengths:**

* Ambitious goal of optimizing the a stock portfolio.

**Weaknesses:**

* The paper does not clearly define what type of behaviors it expect to generate, and why would these behaviors be useful for the stock portfolio. Are they like brokers hired by the owner? What does it mean for two brokers of an owner to compete or to collaborate? Aren't they sharing all the information?
* The figure 1 describing the main concept of the algorithm has only one agent.
* It is not clear how the A2C, PPO and DDPG algorithms had been used in the comparison study, as only the original papers are cited.

**Questions:**

* Please clarify the claimed contributions of the paper in terms of measurable or provable quantities.

---

> ### Author Response · Authors · 2023-11-23
>
> In the new version, we have made improvements to the literature review and model mechanism explanation sections, and changes have been marked in red in the main text.
>
> The subjects of cooperation and competition are the four models trained with s1-s4. There is no data interaction between them, except that each round selects the model with the best reward in the simulation to provide actions for training ddpg. However, they share an environment and there is indirect data interaction.

---

> > ### Comment · Reviewer_uHcE · 2023-11-23
> >
> > Thank you for the response, which, however, does not change my rating.

---

### Meta-Review · Area_Chair_Ldzw · 2023-12-05

**Metareview:**

The reviewers are in agreement regarding the rejection adjudication, and the author's response was insufficient to address their concerns.

**Justification For Why Not Higher Score:**

NA

**Justification For Why Not Lower Score:**

NA

---

### Decision · Program_Chairs · 2024-01-16

Reject